# Could Greater Time Spent Displaying Waking Inactivity in the Home Environment Be a Marker for a Depression-Like State in the Domestic Dog?

**DOI:** 10.3390/ani9070420

**Published:** 2019-07-05

**Authors:** Naomi D. Harvey, Alexandra Moesta, Sarah Kappel, Chanakarn Wongsaengchan, Hannah Harris, Peter J. Craigon, Carole Fureix

**Affiliations:** 1School of Veterinary Medicine and Science, University of Nottingham, Sutton Bonington LE12 5RD, UK; 2Royal Canin Research Center, 30470 Aimargues, France; 3School of Biological and Marine Sciences, University of Plymouth, Plymouth PL4 8AA, UK

**Keywords:** kennelled dog, depression-like state, waking inactivity, anhedonia, affective-state, qualitative behaviour assessment

## Abstract

**Simple Summary:**

Stressed pet dogs, such as when deprived of their owners or after the loss of a social companion, can become inactive and unresponsive. Dogs in this condition are commonly referred to as being “depressed”, but this remains an untested hypothesis. One hallmark of human clinical depression is anhedonia—a reduction in the experience of pleasure. Here we tested the hypothesis that shelter dogs that spend greater time inactive “awake but motionless” (ABM) in their home-pen would also show signs of anhedonia, as tested by reduced responses to a treat filled Kong^TM^. We also explored whether dogs being rated by experts as disinterested in the Kong^TM^ would spend greater time ABM (experts did not know the dogs’ actual inactivity levels). Fifty-seven dogs from 7 shelters were tested in total. Dogs relinquished by their owners spent more time ABM than strays or legal cases, and one association was found between the ABM and the dogs’ response to the filled Kong^TM^, which was in the opposite direction that expected, so does not support the hypothesis that waking inactivity indicates a depression-like state in dogs. Dogs rated by experts as “depressed” and “bored” when exposed to the Kong^TM^, however, spent greater time ABM; we discuss whether ABM could tentatively indicate “boredom” in dogs.

**Abstract:**

Dogs exposed to aversive events can become inactive and unresponsive and are commonly referred to as being “depressed”, but this association remains to be tested. We investigated whether shelter dogs spending greater time inactive “awake but motionless” (ABM) in their home-pen show anhedonia (the core reduction of pleasure reported in depression), as tested by reduced interest in, and consumption of, palatable food (Kong^TM^ test). We also explored whether dogs being qualitatively perceived by experts as disinterested in the food would spend greater time ABM (experts blind to actual inactivity levels). Following sample size estimations and qualitative behaviour analysis (*n* = 14 pilot dogs), forty-three dogs (6 shelters, 22F:21M) were included in the main study. Dogs relinquished by their owners spent more time ABM than strays or legal cases (F = 8.09, *p* = 0.032). One significant positive association was found between the Kong^TM^ measure for average length of Kong^TM^ bout and ABM, when length of stay in the shelter was accounted for as a confounder (F = 3.66, *p* = 0.035). Time spent ABM also correlated with scores for “depressed” and “bored” in the qualitative results, indirectly suggesting that experts associate greater waking inactivity with negative emotional states. The hypothesis that ABM reflects a depression-like syndrome is not supported; we discuss how results might tentatively support a “boredom-like” state and further research directions.

## 1. Introduction

Pet dogs exposed to aversive events, such as when deprived of their owners or after the loss of a social companion, can become profoundly inactive and unresponsive [1], as can laboratory dogs that “give up” and develop “helplessness” (lack of reaction) in response to inescapable stressors [2,3]. Increased inactivity in dogs is also commonly referred to as indicating that a dog is “depressed” or showing signs of “depressive-like behaviour”, both in the public domain [4,5] and peer reviewed literature [6,7,8,9]. However, since the seminal demonstration of “helplessness” in dogs by Seligman and colleagues [2,3], no empirical evidence has been collected to date specifically to investigate the possibility that dogs could display other depression-like symptoms.

In humans, clinical depression (by which we mean *major depressive disorder* or *depressive episodes* to encompass, respectively, Diagnostic and Statistical Manual of Mental Disorders 5 (DSM-V) [10] and International Statistical Classification of Diseases and Related Health Problems (ICD-10) [11] terminologies) is a debilitating mental illness that is commonly triggered by chronic stress, especially in individuals with a predisposition to developing the condition due to innate genetics or aversive early life experiences [10,12,13,14,15]. Various cognitive changes are hypothesised to be involved in both its aetiology and maintenance [16,17,18,19]. One such cognitive change is pessimistic judgements bias, where individuals experiencing a low mood make more negative judgements about ambiguous situations [17,18]. A second is “learned helplessness”, where an individual comes to believe that desired outcomes are improbable and aversive outcomes likely, that no action on the part of the individual can alter this, and as such stop acting [2,16]. 

Diagnosis of clinical depression is based on the co-occurrence of a suite of cognitive, affective and behavioural clinical signs, present for several weeks, and interfering with abilities to cope with everyday life. Such signs include a persistent low, sad mood, anhedonia (a reduction in pleasure), changes in sleep and appetite patterns, difficulty in concentrating and taking decision, and fatigue [10,11]. Importantly, clinically depressed patients often show increased inactivity [10,17] that can take many forms, including not partaking in activities they once enjoyed, or not doing chores that need to be done [18], reduced physical activity [19,20], and difficulty taking part in or initiating social activities [10,17,21]. 

Whether inactive dogs, or at least those displaying certain forms of inactivity in certain contexts, are in depression-like states remains an untested, and yet plausible hypothesis. Most of the properties described in clinically depressed patients might indeed not be unique to humans [22,23,24], and biomedical scientists have been modelling some of the illness symptoms in non-human animals for decades [14,25,26,27]. Because the aetiology of human depression emphasizes aversive life events and chronic stress as common triggers [10,12,13,14,15], increased inactivity in dogs, in tandem with chronic stress or traumatic events, could potentially be indicative of a depression-like condition. Moreover, learned helplessness, one of the cognitive features of clinical depression described above, has also been shown in dogs, and is a phenomenon typically accompanied by an overall decrease in activity [28]. Again, because the aetiology corresponds to cognitive theories of human depression [16,29,30], this inactivity is believed to be a depression-like behaviour. Furthermore, specific forms of waking inactivity displayed in the home environment in other mammalian species have been shown to be associated with key diagnostic features of human depression—anhedonia in riding horses [23] and “helpless” responses in laboratory mice [31].

Do the above-mentioned studies demonstrate that greater level of waking inactivity in the home environment reflect a depression-like state in the domestic dog? They do not provide sufficient evidence. However, altogether they make testing this hypothesis very worthwhile, considering the negative implications that depression-like states would have for dogs. Providing empirical evidence for the existence of such a condition in domestic dogs would be the first step towards developing evidence-based treatments for these animals, and to put these behavioural problems into focus as a legitimate aspect of clinical animal behaviour. Such an understanding of the behaviour of companion animals could lead to improvements in their welfare and management, and potentially even improve the quality of the dog-human relationship for individuals whose behaviour may have gone otherwise unrecognised and misunderstood. Moreover, if shown that dogs can be in a state analogous to human depression (and methods of identifying such a state can be validated), this could help to bring behavioural endpoints for animal research of affected dogs. Besides, whilst animals in naturally occurring depression-like states may represent improved models of human depression, it could potentially invalidate other research where stress and depression are not of interest [32,33]. Using dogs in depression-like states would, therefore, invalidate the results of research projects, implying wastage of animals and violation of the 3Rs’ Reduction [34].

Kennelled dogs, such as those housed in research facilities or rescue shelters, are exposed to an array of chronic stressors including minimal exercise, lack of positive social interactions, disrupted routines, high noise levels and a lack of control over their environment [2,35]. For shelter dogs in particular, such situations could be exacerbated by the abrupt loss of their previous owners, with whom they may have formed strong attachments [1]. There is ample evidence that kennelling can have negative impacts on dogs welfare, however there are currently a lack of methods for measuring their emotional welfare [36]. Because of the chronic stressors and potential traumatic events dogs in rescue shelters are exposed to, we propose that shelter dogs are a suitable model for testing the hypothesis that greater time spent displaying waking inactivity could reflect a depression-like condition in kennelled domestic dogs.

We tested this hypothesis by investigating the association between greater time spent inactive “awake but motionless” in the home-pen and a core symptom of human clinical depression—anhedonia. Anhedonia has been successfully modelled in biomedical studies in rodents, primarily via inducing and recording reductions in sucrose intake [37]. Evidence that sucrose ingestion is pleasure-driven includes that rodents will eat sugar even when fully sated [38] and that it involves the same opioid-mediated reward pathways as sexual behaviour and some recreational drugs [38]. Furthermore, reduced sucrose intake by rodents is induced by chronic stressors [37], alleviated by anti-depressant drugs [26], and co-varies with other depression-like features, including learned helplessness [39] and negative judgement biases [40]. Dogs are equipped with sweet receptors [41], directly translating the paradigm commonly used in laboratory rodents (i.e., comparing the amount of diluted sucrose solution vs. pure water consumed), however appeared practically challenging in rescue shelters. It requires a long exposure (usually several hours) to the solutions, as well as an adequate delivery mechanism—some liquid might be “wasted” if the dog paws at a drinking bowl (biasing measurement of liquids consumed)—and pure sucrose intake might be perceived as an unhealthy diet by shelter staff which might reduce willingness to participate in the study. Therefore, we chose to assess interest for, and consumption of, palatable solid “treat” foods commonly used as dog training aids (i.e., that dogs are motivated to work to get access to) as a proxy for anhedonia. We predicted that dogs that spend greater time awake but motionless in their home-pen would also show a reduced interest in, or consumption of, the palatable treat food. 

As mentioned at the start of this introduction, greater level of waking inactivity in dogs is commonly referred to in the public domain and professionals working with dogs as indicating the dog being “depressed”, although to date there is no empirical evidence supporting this interpretation. Our second aim was, therefore, to explore, using qualitative behavioural assessment (QBA) methods, whether dogs being perceived by a group of experts as apparently disinterested during the food test would also spend greater time awake but motionless in their home-pen (the experts being blind to the dogs’ actual home-pen inactivity levels). 

## 2. Materials and Methods 

The study complied with the European Communities Council Directive of 24 November 1986 (86/609/EEC) and was approved by the University of Bristol Animal Welfare Ethical Review Board in January 2016 (UB/15/072). Permission to approach RSPCA (Royal Society for the Prevention of Cruelty to Animals) shelters was obtained from the Head of Companion Animals Department, the Chief Veterinary Officer and the Chief Scientific Officer in July 2016. Dog husbandry and care were under the management of the shelter staff.

### 2.1. Subjects

This study was conducted in two parts: a pilot phase, where preliminary data were collected, analysed and used to conduct a sample size estimation (see Appendix A section) as well as qualitative behavioural analysis (QBA) and a full study on an independent sample to meet the sample size requirement. The inclusion criteria for selecting dogs that could participate in both parts of this study were four-fold. First, dogs must not have an existing health condition (as diagnosed by a qualified veterinarian based on physical examination). Second, dogs must be aged between 12 months and 10 years of age. Third, dogs must not be on a reduced calorie diet (which could invalidate anhedonia food consumption testing through confounding impacts upon motivation to eat). Fourth, dogs must have been housed in the shelter for a minimum of 1 week at the time of video observation (behaviour of shelter dogs has been shown to become repeatable or stable after 1 week in the shelter [42]). 

We recruited a total of seven shelters in the United Kingdom (either RSPCA or private shelters), in which observations were carried from March 2016 to December 2017 (*shelter 1*: March, June, September 2016 and January 2017; *shelter 2*: August 2016; *shelter 3*: October 2016, February 2017; *shelter 4*: February 2017; *shelter 5*: October 2017; *shelter 6*: November 2017; *shelter 7*: December 2017). Ninety dogs originally met the inclusion criteria, of which 33 had to be excluded from analyses due to being rehomed, developing health problems over the course of the study, or due to recording equipment failures. In all shelters, the dogs were individually housed in two-compartment kennels, entirely cleaned once a day. All dogs were fed twice a day (approximately 08:30 and 16:00–17:00), but two animals were fed an extra meal around lunchtime. Water was provided ad libitum. In all shelters, the dogs were provided daily with a Kong^TM^ either around lunch time or around 16:30–17:00 as part of their normal management routine. In all shelters, all dogs were walked twice a day (once for 10 and once for 20 min) by shelter staff or volunteers. 

#### 2.1.1. Quantitative Analyses Subject Demographics (N = 43)

Complete datasets were successfully collected for 43 dogs in the full study (22 female, 21 male), from across six shelters (*shelter 1*: 8 dogs; *shelter 3*: 7 dogs; *shelter 4*: 5 dogs; *shelter 5*: 10 dogs; *shelter 6*: 9 dogs; *shelter 7*: 4 dogs; see Table 1). Fifty-eight percent (25 dogs) of these were neutered, and 44% (19 dogs) were classified according to the American Kennel Club groupings as being from a working/herding/sporting breed; this binary classification allowed us to include crossbred dogs in the analysis; where parent breeds were stated they were assigned a 1 or 0 accordingly, or when no breed was stated (mixed) dogs were assigned a 0 as they could not be classified into this breed group. In total, 49% (21 dogs) were voluntarily relinquished, 16% (7 dogs) were found as strays, and 33% (14 dogs) were seized as part of legal cases (plus one dog missing origin data). The mean age of the dogs was 4.01 years (SD ± 2.17), with the youngest being 1 year and the oldest being 10 years of age. Excluding an outlying long-stay dog who had spent 210 weeks in the shelter, the mean number of weeks spent in the shelter was 8.1 (SD ± 8.0), with the minimum number being 1.4 weeks and the maximum being 45.4 weeks.

#### 2.1.2. Qualitative Behaviour Analyses (QBA) Subject Demographics (N = 14)

A total of 14 dogs (7F: 7M) from three shelters, recruited during the pilot study, were utilised here (*shelter 1*: 7 dogs; *shelter 2*: 2 dogs; *shelter 3*: 5 dogs) (see Table 1). QBA analyses were performed on the dogs from the pilot sample, as the complete datasets from the 43 other dogs described above were not available at the time we could organise the QBA assessment. Dogs were aged 1–9 years (average 3.02 ± 2.26 years), and 43% (6 dogs) were classified according to the American Kennel Club groupings as being from a working/herding/sporting breed. In total, 36% (5 dogs) were voluntarily relinquished, 21% (3 dogs) were found as strays, and 29% (4 dogs) were seized as part of legal cases (plus two dogs with missing origin data). The mean number of weeks spent in the shelter was 8.4 weeks (SD ± 7.7), from 3.4 weeks to 15.6 weeks.

### 2.2. Home-Pen Activity Budget

Dogs were video recorded in their home-pens for a total of 6 h, blocked over three days and three 2-h time periods. The three 2-h recording periods were classified as: AM, between 09:00 and 11:00, early PM (EPM), between 11:30 and 13:30, and late PM (LPM), between 14:00 and 16:00. Each dog (maximum 9 dogs studied simultaneously in a 3-day recording block due to camera availability) was recorded for one 2-h period each day following a day and period blocked design, so that each dog had 2-h of footage from each period.

Two GoPro Hero 3 (white edition) cameras were used per dog, positioned outside (therefore not reachable to the dog) at either end of the kennels using specially made wooden mounts and GorillaPod^®^ for mesh kennel walls, and GoPro glass mounts for kennels with solid walls and glass door panels. The cameras were placed at the height of the dog’s head and angled inwards to capture the maximum range of the kennel floor space. The mounts were left in place for the full three-day period, whilst the cameras were removed after each observation period to be recharged and for video file extraction.

An ethogram (Table 2) was developed based upon previous published work [23,31,35,44,45,46,47] to record the dog’s behaviour in the home-pen. Behaviour was sampled via instantaneous sampling [48]. “Not visible” was selected if the dog was entirely not visible in the camera shot (i.e., was not in the kennel) or if the view was so obscured that identifying its behaviour became ambiguous. For the behaviours being characterised by either a lack of movement (e.g., awake but motionless, sleeping) or repetition (abnormal repetitive behaviours), 10 s of footage was watched continuously 5 s either side of the scan point in order to best determine the correct following action [31,49]. The behaviour we hypothesised to reflect a depression-like condition in dogs, being awake but motionless (ABM) was defined as follows (adapted from previous studies [23,31]): “*The dog is completely motionless (no head, body or ear movements) with eyes open apparently staring (anywhere). Dog may be lying, sitting or standing but not vocalizing. If sitting, head may be in a “drooped” position with head lower than or level with their spine. State must last for at least 5 s*”.

Videos were scored using the Behavioural Observation Research Interactive Software (BORIS) [50] by trained observers (C.F., C.W., H.H., P.J.C., and an intern, Miss S. Vuillermet). Observers were all blind to the dogs’ results in the anhedonia test at the time they were scoring home-pen behaviour. Training of observers consisted first of watching videos together with C.F. or N.D.H. to identify examples of each item from the ethogram. Following this, C.F. and the observers worked independently to scan 15 min of footage each (scanning each minute), from four dogs gathered during the refinement stage of the pilot. The scans were visually checked for agreement, with any non-matching scans reviewed and discussed until the observers agreed on the code to be assigned. This was repeated with a series of random 15-min samples for each dog until raters assigned the same code for all scans on two consecutive occasions. Agreement was re-checked once (approximately around mid-term data extraction) for the pilot study and it remained excellent (only one scan differed, therefore we did not evaluate the difference statistically), or at any time requested by any of the observers on specific data point.

In order to be time efficient, but not lose accuracy of the data, it was necessary to identify the maximum sample interval we could use that would produce a representative activity budget. Shorter sample intervals form more accurate representations of behaviour but are less time and cost effective than longer intervals [51]. For the first seven dogs from Shelter A scans were taken every 30 s over 4 h (480 scans) in total. An activity budget was calculated for each dog comprising the proportion of scans seen exhibiting each point event from the ethogram. This was done iteratively for all seven dogs for the original 30-s interval scans, then utilising every second scan (representing 1-min intervals), every third scan (representing 1.5-min intervals) and every fourth scan (representing 2-min intervals), creating 4 different datasets. The mean proportion of scans spent in each behavioural state was calculated for each dataset, followed by the difference in the mean between the 30-s reference data and each of the second, third and fourth scan data (the Error Proportion; EP). If the EP for the larger interval datasets for any behavioural state was less than 10% different from of the 30-s mean estimate then it was considered to have retained accuracy [51]. In this way, the longest interval that produced mean behavioural estimates most similar to the 30-s reference sample, for the greatest number of behavioural states, was selected for subsequent video analysis (see Appendix A for a full description of this process). Intervals of 1.5-min were deemed to produce acceptably accurate time budget estimates and were utilised for all subsequent home-pen video analyses.

### 2.3. Anhedonia Test

A stuffed Kong^TM^ toy (KONG Company, Golden, CO, USA) was placed on the floor of the dogs pen (small, medium, large or extra-large Kong^TM^ assigned according to dog size as outlined here [52]. The toy was stuffed with a mix of the dog’s own standard dried biscuits, soaked dried chicken pieces (a common dog training aid) with KONG Stuff’n Paste to bind and the large hole was sealed with a 3-cm long piece of hot dog sausage. Ingredients were mixed following the ratio of 2:3 for biscuits to soaked dried chicken pieces, for each 2 “squirts” of Stuff’n Paste, in order to keep the Kong^TM^ effectively filled the same. All dogs were already habituated to the Kong^TM^ toys (provided daily in each of the recruited shelters) and it has been shown that dogs habituated to Kong^TM^ treat filled Kong’s as feeding devices (i.e., dogs behave in a way demonstrating that they expect to get food from it) as opposed to a rubber toy [53], which could have confounded the results. To prevent potential neophobic reactions to the Kong^TM^ contents, dogs were offered a small amount of the foods used on the day prior to the test to ensure willingness to eat all of them. To limit the impact of appetite, dogs were given the Kong^TM^ between 15 to 30 min after they had consumed their normal ration of dog food, as hedonically motivated behaviours seem more driven by opportunism and external stimuli (i.e., eliciting cues such as odours) than by states of deprivation [23,54]. To control further for motivation for food, the dogs’ latency to approach their normal ration of dog food (less palatable than treat food used to stuff the Kong^TM^) was recorded once for each dog on the day previous to the Kong^TM^ consumption test. The dog’s behaviour was video recorded using the same setup as for the home pen recordings but was conducted on the day after the home pen recordings were completed. Following 30 min of exposure, the Kong^TM^ toys were removed from the pen. The filled toys were weighed before being given to each dog and weighed again after to enable calculation of how much food mix was consumed.

N.D.H. and two research assistants (H.H. and Dr G. Miguel-Pacheco), blind to the dogs’ activity budgets, used the video footage of the consumption test to subsequently score: the total time the dog spent interacting with the food toy (defined as paw or muzzle in contact with, or sniffing the Kong^TM^, including time stood chewing the food mix but not in physical contact), expressed as a proportion of the total test duration (*Kong_Prop_Time*); the number of bouts of interaction with the Kong^TM^ (a bout was considered to have ended when a dog ceased to physically contact the Kong^TM^ and ceased to chew food retrieved from the Kong^TM^, and began again when the dog re-initiated contact with the Kong^TM^; *Kong_Bout_N*), and the duration of each bout (averaged for each dog; *Kong_Av_Bout_Time*). Data regarding the percentage of the food mix eaten by the end of the 30 min test (*% of Kong Eaten*) was included as a fourth test variable.

### 2.4. Qualitative Behaviour Assessment

QBA focuses on assessing observers’ interpretations of an animal’s behaviour (how does the dog appear to feel) rather than the observers quantifying individual behaviours themselves. Following developed methodologies, a QBA term list was utilised from a previously validated list designed to assess shelter dog behaviour [55]. The original 20 term list was reduced to 17 terms for our purposes, as three terms were not relevant to the footage we had as they related to social behaviour (aggressive, attention-seeking and sociable) and all dogs were housed alone. Data collection was conducted by N.D.H. and A.M. and took place at Waltham on September 1, 2017. A total of 6 participants, all experienced in working with dogs, were recruited (including A.M.). Participants were briefed on the project and trained in QBA using training clips (of dogs not in the study) by N.D.H., prior to scoring. The QBA terms were discussed and the definitions clarified as a group until a consensus was reached on how each term would be defined. A total of 5 clips were scored per dog, for 14 dogs (see Table 1 and Section 2.1.2), with each clip being 30 s long and selected to be evenly spread across the Kong^TM^ test period. Ten minutes were allotted to score each dog (2 min per clip in total), scoring three dogs each time before a break to help prevent people becoming mentally fatigued and keep scoring quality high. All participants were blind to the home-pen conditions and the quantitative Kong^TM^ data extraction results.

### 2.5. Statistical Analysis

#### 2.5.1. Quantitative Analysis

Statistical analysis was conducted using SPSS^®^ v. 22 (SPSS Inc., Chicago, IL, USA). Descriptive statistics were calculated to summarise the behavioural variables observed during the home-pen and anhedonia tests. Time spent ABM and model residuals were not normally distributed, so ABM was transformed into a logarithmic scale (after adding 1 to remove zeros); the logarithmically transformed variable is indicated with “lgABM”. Breed could not be included individually in the models, so dogs were classified according to the American Kennel Club groupings as being from a working/herding/sporting breed or not, which allowed for crossbreeds to be included. Univariate linear regression models were utilised to investigate potential associations between time spent ABM (dependent variable) and each independent variable: sex; neuter status; shelter; age; origin; video observer; American Kennel Club working/herding/sporting breed (yes/no); and each anhedonia Kong^TM^ test variable. The main outcome variable for this study was the dog’s response to the Kong^TM^, therefore effects of all other variables were investigated as potential confounding variables. These variables (age, weeks in shelter, neuter status, female, weight, and working/herding/sporting breed group) were tested for associations amongst each other using Kendall’s tau-b correlations (in case of ties) and were included in multivariate models together where associations were found. Model residuals were visually assessed for normality, and collinearity in multivariate models was checked for and ruled out using variance inflation factor (VIF) statistics.

#### 2.5.2. QBA Analysis

Data was analysed according to accepted QBA methodology alongside original data on ABM and Kong^TM^ interactions [56]. Principal components analysis (PCA) (unrotated and based on a correlation matrix) was used to analyse the data using QBA scores from each of the 5 clips initially to test for an effect of clip on reliability. The Kendall Correlation Coefficient (Kendall’s W) was used to evaluate inter-rater reliability of the resulting PCA scores. Kendall’s W varies from 0, indicating no agreement, to 1, indicating perfect agreement, and values greater than 0.5 were considered acceptable. Following this, the QBA data for all clips and all raters were combined into a final PCA along with the home-pen ABM, and anhedonia test data for each dog.

## 3. Results

### 3.1. Quantitative Results

Overall, dogs were most frequently recorded observing (median time: 37.0%), walking (10.5%), lying down with head or ears mobile (9.9%) and in behavioural or postural transition (8.6%) in their home pen (Table 3). Crucially, the dogs did display the behaviour we hypothesised to reflect a depression-like condition (being awake but motionless “ABM”), for a median time of 3.1% of the scans (first quartile 1.2%, third quartile 6.8%) with clear variation between individual dogs (from 0 to 20.4% of scans). This ranks similarly to the pilot group of dogs, for whom ABM was the sixth most common behavioural category exhibited and spent a median time of 2.5% and a maximum of 16.1% of time in this state (see Appendix A section).

There were no significant differences in (lg)ABM between videos coded by the four different observers (ANOVA (analysis of variance) with Tukey post-hoc: F = 0.62, *p* = 0.605) nor between the shelters (ANOVA with Tukey post-hoc: F = 0.41, *p* = 0.801) or age (linear regression: t = 0.199, *p* = 0.843). Further, no significant differences were found in (lg)ABM according to sex or neuter status (ANOVA with Tukey post-hoc: F = 0.776, *p* = 0.514), nor between dogs’ breed type (American Kennel Club groupings) (ANOVA with Tukey post-hoc: F = 2.07, *p* = 0.158). However, dogs that were relinquished to the shelter by their owners spent significantly more time ABM than dogs that were seized as part of welfare cases (controlling for time spent in the shelter, Figure 1; ANOVA with Tukey post-hoc: F = 8.09, *p* = 0.032).

With regards to the anhedonia test, individuals varied in the time they spent interacting with the Kong^TM^, spending a median time of 29.0% of the test interacting with the Kong^TM^ (first quartile 14%, third quartile 48%, range from 0% to 95%). The manner in which dogs interacted with the Kong^TM^ also differed in terms of how many bouts of interaction they engaged in—dogs engaged for a median number of 8 bouts of interaction, with a minimum of 1 and maximum of 28 bouts (first quartile 3, third quartile 12). Inter-individual variation was also observed in the proportion of the mix the dogs ate from the Kong^TM^, from 0 (none) to 100% (median 96.4%, first quartile 38.3, third quartile 100.0).

The variables latency to eat the mix in the Kong^TM^ (a control for potential neophobia), and latency to start eating their regular meal (a control for general interest in food) showed very little variation, with most dogs (93% and 88%, respectively) starting to eat within a few seconds (on average 1.87s SD ± 2.3 for trying the new mix, and 1.88 s SD ± 1.4 for regular meal). For this reason, these two variables were not included in these analyses. Although origin was associated with ABM, it was not considered to be something that should be “controlled” for, as it could be a factor associated with the likelihood for developing a negative affective response to being in a shelter.

Significant correlations were found between the factors female, weight, and working/herding/sporting breed group (Appendix A), so these were included together in any multi-variate models, with collinearity checked for using VIF diagnostics. Each Kong^TM^ test variable was tested against (lg)ABM in its own regression model with potential confounders (age, weeks in shelter, neuter status and female, weight, and working/herding/sporting breed group) checked sequentially for stratifying interactions. Only one model was found to be significantly associated with ABM. A model including average bout length of interaction with the Kong^TM^ (in seconds) with the number of weeks spent in the shelter was significant at the *p* < 0.05 level (Table 4). Both variables were positively associated with (lg)ABM, with longer average bout lengths tending to be associated with greater time spent ABM with increasing time spent in the shelter. This association is in the opposite direction than would be predicted if time spent ABM was an indicator that some of these dogs were in a depression-like state.

### 3.2. QBA Results

Two dimensions were formed from a PCA of the QBA scores: Component 1—Stressed/Anxious to Comfortable/Relaxed, which explained 25.3% of total variance; and Component 2—Interested/Explorative to Bored/Depressed, which explained 15.5% of all variance. Kendall’s W was applied to the PCA scores for each of the 5 clips (to check whether clip order affected the reliability) and averages to evaluate inter-rater reliability. There was no effect of clip and both component scores were deemed suitably reliable, achieving average W-statistics of 0.63 (*p* < 0.001) for Component 1 and 0.53 (*p* = 0.003) for Component 2.

The data for all clips and all raters were then combined in a PCA with the ethogram data ABM, proportion of test interacting with Kong, the number of Kong^TM^ bouts and the average duration of these bouts to place these measures within the QBA component structure (Figure 2, Appendix A). The home-pen score, ABM and the average bout length of Kong^TM^ interaction loaded with the QBA scores “Depressed” and “Bored”, along the axis for Component 1 (meaning that dogs rated as “depressed” or “bored” during the Kong^TM^ test were displaying greater time awake but motionless in their home-pen), whilst the Kong^TM^ scores for proportion of test time spent interacting with Kong^TM^ and the total number of Kong^TM^ bouts loaded towards the Interested/Explorative end of Component 1 (meaning that dogs rated as Interested/Explorative were spending more time and more bouts interacting with the Kong^TM^).

## 4. Discussion

The goal of the study was to test the hypothesis that greater time spent displaying waking inactivity in the home environment could reflect a depression-like condition in kennelled domestic dogs. We tested this hypothesis in rescue shelter dogs by quantitatively investigating the association between greater time spent inactive “awake but motionless” in the home-pen and anhedonia (a core symptom of human clinical depression), using reduced interest in, and consumption of, palatable food as a proxy for anhedonia. Because greater levels of waking inactivity in dogs is commonly referred to in the public domain and amongst professionals working with dogs as the dog being “depressed”, we also aimed to explore whether dogs being qualitatively rated as less interested in the palatable food test would also spend greater time awake but motionless in their home-pen (with the raters being blind to the dogs’ actual home-pen inactivity levels and food test quantitative results).

Our quantitative results do not support the study hypothesis. One of the measures showed a positive association with time spent ABM, with longer average bout lengths interacting with the food dispenser associated with greater time spent inactive in the home pen with increasing time spent in the shelter, which is in the opposite direction than would be predicted if greater time spent ABM was an indicator that dogs were in a depression-like state. Interestingly however, the results from the QBA study have shown that the time spent awake but motionless in the home-pen correlated with scores for “depressed” and “bored” (which overlapped with each other) in the QBA of the Kong^TM^ test videos, despite the data being completely independent and scored blindly. This result shows that people experienced with dogs observed something in the dogs’ demeanour during the anhedonia test that they associated with a negative emotional state (in this case “depressed” and “bored”), which in turn was associated with waking inactivity independent of the anhedonia test. This implies that waking inactivity is not a “normal” healthy behaviour, but instead is associated with a negative emotional state. That the dogs displaying greater time ABM are “depressed” is not supported by our quantitative observations; however, our results might tentatively support a “boredom” hypothesis.

In humans, boredom is a negative affective state induced by a lack of desired stimulation or behavioural opportunities [57]. While the use of the term “boredom” in animals and its relationship to inactivity still needs validation [57,58,59], Meagher and Mason [60] have proposed an operational definition of such a state based on motivation to obtain stimulation. According to this definition, a “bored” animal would display increased willingness to interact not only with positive (appealing) stimuli, but also with neutral and even slightly aversive situations, as a result of an overall elevated motivation to obtain stimulation; of any kind. That the dogs spending greater time ABM tended to display an increased interest in the filled Kong^TM^ toy (as shown by longer bout average lengths interacting with it) would match with the boredom-like state prediction. Furthermore, Meagher and collaborators have tentatively identified in minks a link between apparent boredom and a specific subtype of inactivity that might (partly) resemble the waking inactivity observed in the dogs here: lying down with the eyes open when undisturbed in the home cage [61] (although another study [62] failed to replicate this result).

That dogs that were relinquished to the shelter by their owners spent significantly more time ABM than legal case dogs might also tentatively support an association between waking inactivity in the home kennel and a boredom-like state. It is indeed possible that being kennelled represents for these dogs an even more impoverished environment than it does for dogs from a different origin, as relinquished dogs may have been used to getting various stimulation and positive human contact in the home. Such a loss of enrichment, where stimulation is lacking but individuals remains generally motivated to be stimulated, can trigger boredom [59]. One may, thus, hypothesize that greater levels of waking inactivity in dogs relinquished by their owners, and as such putatively exposed to greater environmental enrichment loss compared to dogs from a different origin; might reflect a boredom-like reaction.

Such a tentative relationship between waking inactivity in the home environment and boredom-like states in dogs remains to be investigated further however, as direct positive association between greater time spent ABM and the positive stimulus was observed only for one measure from the Kong^TM^ test. However, “average bout length” is the most discerning of all the Kong^TM^ variables, as dogs with the same overall time could have very different patterns of interaction in terms of bout numbers, and vice versa, which may be why it was the only significant variable. Crucially however, the dogs’ reactions to neutral and slightly aversive stimuli were not tested here, which are part of the operational definition of boredom-like states in non-humans [60]. Although the current results do not support the depression-like condition hypothesis, we also believe that further research should be conducted into the relationship between waking inactivity in the home environment and depression-like state in dogs, as methodological refinements and complementary investigations are required before it is possible to safely reject this hypothesis. As such, we will discuss further research directions into depression-like hypotheses, which do not mutually exclude conducting more in-depth investigation into boredom hypotheses.

In regards to the assessment of anhedonia, we chose here to assess this phenomenon via reduced interest in, and consumption of, palatable “treat” foods as a proxy, because reduced sucrose-ingestion has been validated in laboratory rodents as a proxy of anhedonia [37]. This approach has several drawbacks, however. First, anhedonia in clinical depression refers to a “markedly diminished pleasure in all, or almost all, activities” [10], and reduced interest in palatable food (that dogs are motivated to work to get access to) does not demonstrate generalised anhedonia. A broader range of activity motivated by positive affect should, thus, be investigated. For instance, our original research plan also included assessing reduced interest and eagerness to play, both self-play interacting with toys and with a person [63]. Within this study context, however, implementation of the play protocols in situ proved challenging for practical reasons, which included difficulties in finding testing areas that were relatively similar across shelters to perform the play with a person protocol, as well as large variation in the dogs’ exposure to toys and play (which could have induced neophobic reactions had we suddenly added toys to the kennels, or conversely increased salience and interest in those dogs not provided with toys, therefore acting as potential confounds when interpreting the self-play with toys in the kennel results). A second drawback is that using solid “treat” palatable food does not directly translate the paradigm used in laboratory rodents (i.e., comparing the amount of diluted sucrose solution vs. pure water consumed [37]. Moreover, as discussed in a previous study [64], low concentrations of sucrose (such as with a strongly diluted solution) might be more sensitive when measuring anhedonia, while solutions or food with significantly more concentrated palatable compounds might be more consumed by stressed individuals, including depressed people, as comfort food intake and carbohydrate “craving” responses. As mentioned in the introduction, our choice of using reduced interest in, and consumption of, palatable “treat” foods as a proxy of anhedonia was motivated by feasibility reasons within the available time frame of this study. Longer stays in shelters for future research or involvement of laboratory dogs from research centres environments should allow tackling these practical concerns, e.g., allowing for more direct translation of validated paradigms (longer measurements, with adapted drinking apparatus) and assessment of a broader range of activity motivated by positive affect. Further research would also include assessing the co-variation of ABM not only with anhedonia but also with other key symptoms of depression, as discussed in previous studies [22,23], as well as into inter-individual variations in susceptibility to developing depression-like states, such as breed differences and active and passive responder styles.

Along with assessing anhedonia and the existence of other symptoms of depression more fully, further research (including ours) would benefit from refined definitions or measurement of inactive behaviour(s) relevant to hypothesis(es) under evaluation. Indeed, while inactive behaviour in shelter dogs is mentioned in published papers, to date we could not identify with confidence reports of ABM in the scientific literature. For instance, dogs have been reported as “resting” when “lying down with eyes open *or closed*” [45] (our emphasis in italic), or with “abdomen touching the ground with its dorsal, caudal or lateral side, whilst legs are extended forwards, curled close to the body or laid to one side; eyes are open” [65], which compared with ABM does not appear to exclude the possibility of head or ear movement. Perhaps the closest behavioural description we found comes from a study [66] describing “passive gazing” as the dog being “still and its eyes are open, but its attention does not appear to be focused on anything in particular”. That study, however, investigated how kennel sizes influences the dogs’ behaviour (finding no effect on this particular behaviour) and did not study the possibility that this behaviour could reflect a depression-like condition. This diversity in the way forms of inactivity are defined highlights a broader issue within the field of inactivity-related investigations, i.e., that inactivity is often considered simply a default state, or not associated with specific hypothesis, rather than a true “behaviour” [58] and is differentially defined between studies, therefore limiting cross-study comparisons.

Lastly, we built our hypothesis and chose to define here the specific form of inactivity ABM following previous works showing that being awake with eyes open, motionless in the home environment was associated with key diagnostic features of human depression in several other mammalian species: horses, laboratory mice and non-human primates [23,31,67]. In the current study, we defined waking inactivity as follows: “*Dog is completely motionless (no head, body or ear movements) with eyes open apparently staring (anywhere). Dog may be lying, sitting or standing but not vocalizing. If sitting, head may be in a “drooped” position with head lower than or level with their spine. State must last for at least 5 s*”. Despite general similarity, there were variations from above-mentioned studies in the way we defined the inactive state of interest. First, postural criteria included in our dog definition were less specific than for instance the flat head or back alignment included in the criteria for being “withdrawn” in horses [23] or the “crumpled” body posture of “depressed” monkeys [24,67]. The “drooped” posture was an optional part of the definition in the dog and was not commonly (if at all) observed. Moreover, we did not have the scope to quantify the (lack of) reactivity of the dogs as we did in horses, for example, in which we measured the animals’ reactivity to a range of visual and tactile stimuli [68]. Lastly, the behavioural method we used for recording the inactive behaviour in the dogs (instantaneous scan sampling with a 5 s interval around the scanning point) was chosen as a compromise between measuring the exact durations of each bout length (that would allow more precise quantification of inter-individual variations) and the time required to extract data from the videos (scan sampling being relatively fast). Continuous recording of the behaviour could have, thus, been more appropriate; its downside being nevertheless that the time required to extract behaviour from the 400 + hours of video recording would have conflicted with the budget available for research assistants to extract the data. For these reasons, it is, therefore, possible that the form of waking inactivity we measured in the current study partly differs from the above-mentioned forms (e.g., is less “profound”), and that greater times displaying that behaviour are either associated with a different affective state (e.g., boredom) or even simply part of the normal behavioural repertoire of kennelled dogs. Further investigations using continuous recording of the behaviour, refining postural aspects of the definition, and comparing times spent displaying ABM in more diverse and enriched environments, including at home, would help in addressing this question.

## 5. Conclusions

Through this study we showed that shelter dogs spend an average (median) of 3.1% of their time “awake but motionless”. Distinct inter-individual variation in this behaviour was present, with some dogs not spending any time “awake but motionless” and others spending 20.4% of their time in this state. Being relinquished to the shelter by the dogs’ owner (as opposed to being seized as part of a welfare case) was associated with greater time spent awake but motionless in the home-kennel. We could not conclude that the most inactive dogs in our study displayed signs of being in a depression-like state, although methodological refinements and complementary investigations are required before it is possible to safely reject this hypothesis. However, our results highlight a potential association between being awake but motionless (apparently doing nothing) in the home-kennel and a boredom-like state in shelter dogs. Boredom is reported to feel highly aversive in humans, and this result opens the door to further investigations of this important concept, taking the first steps towards validating a non-invasive behavioural indicator of “boredom” in the dog. Identifying measures of a boredom-like state will be required if we are to better understand the impact of housing, management and research procedures on kennelled and laboratory animals to maximize their welfare, and ultimately develop treatments for affected individuals.

## Figures and Tables

**Figure 1 animals-09-00420-f001:**
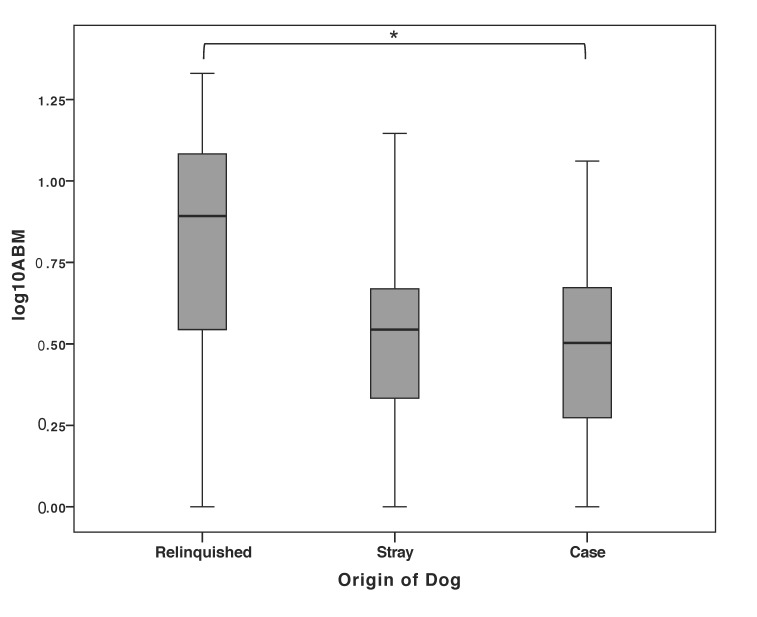
Boxplot showing the distribution (median and interquartile ranges) of time spent awake but motionless (ABM; logarithmically transformed) between dogs that were relinquished by their owners (*n* = 21), found as strays (*n* = 7) or seized as part of animal welfare legal cases (*n* = 14). ANOVA with Tukey post-hoc: F = 8.09, * *p* = 0.032.

**Figure 2 animals-09-00420-f002:**
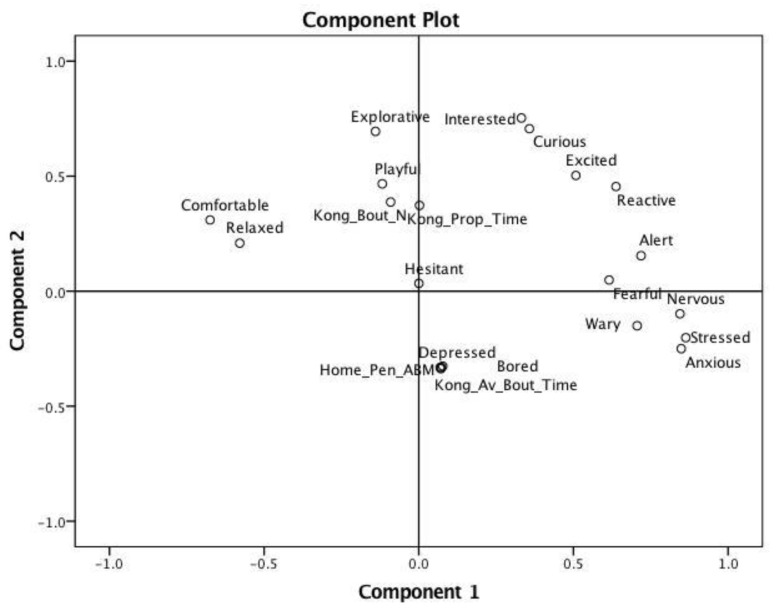
QBA scores overlaid with Kong^TM^ test variables and home-pen awake but motionless ABM behaviour. The axis for Component 1 (25.3% of variance) reflects Stressed/Anxious to Comfortable/Relaxed, whilst the axis for Component 2 (15.5% of variance) reflects scores for Interested/Explorative to Bored/Depressed. When interpreting Figure 2 and Appendix A, it is important to note that all QBA participants (n = 6) were blinded to the dogs’ prior ethogram home-pen and Kong^TM^ test scores.

**Table 1 animals-09-00420-t001:** Individual characteristics of the 14 pilot study dogs included in the qualitative behaviour analysis (QBA) and the 43 dogs from the main study for which complete data sets were obtained.

ID	Shelter	Sex	Intact	Age (years)	Breed	W/H/S breed	Origin	Study
10	1	M	MD	1	Labrador cross	Yes	Rel.	QBA
12	1	F	MD	1	Husky	Yes	Rel.	QBA
14	1	F	MD	2	German shepherd	Yes	Rel.	QBA
15	1	M	MD	2	Mixed	No	Rel.	QBA
18	1	M	MD	1	Lurcher	No	Stray	QBA
19	1	M	MD	2.5	Jack Russell Terrier	No	Stray	QBA
20	1	F	MD	5	ABD x Mastiff x DDB	Yes	Stray	QBA
24	2	F	MD	4.5	SBT	No	Rel.	QBA
26	2	M	MD	2	Boxer cross	Yes	MD	QBA
33	3	F	Yes	5	Springer Spaniel	Yes	Case	QBA
34	3	M	Yes	2	Beagle cross	No	Case	QBA
37	3	M	Yes	1.25	Bichon frise x Toy poodle	No	Case	QBA
39	3	F	Yes	9	SBT	No	Case	QBA
40	3	F	Yes	4	Jack Russell Terrier x SBT	No	MD	QBA
42	1	M	No	6	Border Collie	Yes	Rel.	Main
43	1	M	No	5	Boxer	Yes	Rel.	Main
44	1	F	Yes	6	SBT cross	No	Rel.	Main
45	1	F	No	8	Yorkshire terrier	No	MD	Main
46	1	M	No	2	SBT	No	Rel.	Main
47	1	M	No	3	Husky Hound	Yes	Rel.	Main
48	1	F	No	3	Parson Jack Russel cross	No	Rel.	Main
49	1	M	No	4	Husky Hound (small)	Yes	Rel.	Main
50	3	M	Yes	1	French Bulldog	No	Case	Main
51	3	M	Yes	2	French Bulldog	No	Case	Main
52	3	F	No	2	Rottweiler x GSD x Collie	Yes	Rel.	Main
53	3	M	No	6	SBT x Boxer x Labrador	Yes	Rel.	Main
54	3	F	No	6	Bull Lurcher	No	Rel.	Main
55	3	F	Yes	1	French Bulldog	No	Case	Main
56	3	F	Yes	3	Mixed	No	Case	Main
58	4	F	No	3	Alaskan Malamute cross	Yes	Rel.	Main
59	4	F	No	4	Border collie	Yes	Rel.	Main
60	4	M	No	2	Saluki	No	Rel.	Main
61	4	F	No	4	Lurcher	No	Rel.	Main
62	4	F	No	7	SBT cross	No	Rel.	Main
64	5	M	No	2	GSD x Akita	Yes	Stray	Main
65	5	M	No	5	Mixed	No	Rel.	Main
66	5	M	No	7	SBT	No	Stray	Main
67	5	F	No	2	Mixed	No	Rel.	Main
68	5	M	No	2	Akita	Yes	Stray	Main
69	5	F	Yes	4	Akita	Yes	Stray	Main
70	5	F	No	4	Border collie	Yes	Stray	Main
71	5	M	No	2	Bichon frise	No	Rel.	Main
72	5	F	No	5	SBT	No	Stray	Main
73	5	M	Yes	6	GSD	Yes	Stray	Main
74	6	F	Yes	2	Yorkshire terrier	No	Case	Main
75	6	F	Yes	7	Yorkshire terrier	No	Case	Main
76	6	F	Yes	1	Pug x Bichon frise	No	Case	Main
77	6	F	Yes	4	Chihuahua	No	Case	Main
78	6	M	Yes	10	Bichon fries cross	No	Case	Main
79	6	F	No	6	SBT	No	Rel.	Main
80	6	F	Yes	4	SBT	No	Rel.	Main
81	6	M	Yes	5	Labrador	Yes	Case	Main
82	6	F	Yes	3	Shizu cross	No	Case	Main
83	7	M	Yes	MD	Newfoundland	Yes	Case	Main
84	7	M	No	2	SBT	No	Rel.	Main
86	7	M	Yes	MD	Newfoundland	Yes	Case	Main
87	7	M	Yes	MD	Newfoundland	Yes	Case	Main

MD = Missing data; M = male, F = Female; GSD = German Shepherd Dog; ABD = American Bulldog; DDB = Dogue de Bordeaux; SBT = Staffordshire Bull Terrier; Rel. = Relinquished. Breed information was obtained from pedigrees (when available) or visual inspection (which for the latter might involve some overestimation of Staffordshire Bull Terrier crosses [43]). W/H/S breed indicates whether the breed stated or the parent breeds (if stated) were in a working/herding/sporting breed group according to American Kennel Club classifications. ‘x’ indicates a cross between breeds.

**Table 2 animals-09-00420-t002:** Ethogram for determination of activity budgets of dogs in their home pen (adapted from previous studies [23,31,35,44,45,46,47]). Note that eye invisibility when the dog is scored “Eyes out of sight” does not allow for determination of the correct action (e.g., sleeping, or awake but motionless), which would impair any interpretations of the results; consequently, no subsequent analyses were performed on this variable.

Category	Description
Not visible	Dog is out of the kennel, or out of sight and no options can be confidently selected.
Awake but motionless (ABM)	Dog is completely motionless (no head, body or ear movements) with eyes open apparently staring (anywhere). Dog may be lying, sitting or standing but not vocalizing. If sitting, head may be in a “drooped” position with head lower than or level with their spine. State must last for at least 5 s.
Sleeping	Dog is lying down, motionless with eyes closed. State must last for at least 5 s.
Eyes out of sight	Could be sleeping or ABM but eyes are out of sight.
Observing	Dog is located at the boundary of the kennel, standing or sitting and attention is focused outside of the kennel (i.e., the head or the head and the body are oriented out of “aperture”, mesh or glass door).
Lying down head or ears mobile	Dog is lying down eyes open, no body movement but may be moving ears and/or head, including any short, subtle movements. Dog should not be vocalising.
Walking	Forward motion of the dog with all four legs in motion [includes trotting, fast-paced walking or running.
Jumping	The dogs front two paws, or all four paws, are off the ground; may occur in bouts, or may be still with front two paws on a wall or door.
Sniffing	Nose angled downwards and in close proximity to the floor. Often the head will make sharp side-to-side movements. Can be done while the dog is in motion or stationary.
Urinating/Excreting	Release of urine or faeces onto floor.
Grooming	Licking or chewing of self.
Eating	Dog swallows an item it has in its mouth. Results in sequence of characteristic movements of the mandibular.
Drinking	Series of movements where the dog’s tongue touches the liquid up to the swallow. A dog may often stop in between drinking to breathe. The head performs a bobbing motion.
Interact with Object	Can include: chewing of an object that is not food; touching an object with their front paws repeatedly (pawing); tossing an object in the air; rolling on an object.
Interact with Person	Can include: licking; touching with paws, snout or body; jumping on; sitting on; lying on; or being petted (dog may be lying, standing or sitting).
Abnormal Behaviour	Including: pacing, walking in a repetitive pattern (at least 3 repeats) usually along a boundary; flank sucking (taking skin in mouth and sucking); wall bouncing (dog jumps towards wall and contacts with limbs repeatedly (>3 times); tail chasing (repeated (>3 times) chasing of tail).
Barking	Vocalisation of loud sounds. Head is often elevated and thrown forwards at the moment of the bark. Often in bouts of multiple barks.
Whining	Prolonged high-pitched sound. Mouth may be open or closed.
Howl	Raise muzzle perpendicular to ground and emit a long, drawl out sound through semi-closed jaws.
None of the above	For example, postural transitions (the dog is standing up, or lying down, exactly at the time of scan), yawning

**Table 3 animals-09-00420-t003:** Median percentage of scans, first (Q1) and third (Q3) quartiles and minimum (min) and maximum (max) values for behaviour in the home-pen (n = 43 dogs). The behaviour are ordered from the longest to the shortest median times spent displaying them, and the behaviour we hypothesise to specifically reflect depression is highlighted in bold.

Behaviour	Median	Q1	Q3	Min	Max
Observing	37.0%	21.6%	49.7%	0.0%	68.5%
Walking	10.5%	5.5%	16.7%	1.2%	30.2%
Lying down head or ears mobile	9.9%	3.4%	14.2%	0.0%	48.1%
None of the above	8.6%	4.9%	15.4%	0.0%	42.0%
Eyes out of sight (could be ABM or sleeping)	4.9%	1.2%	12.4%	0.0%	30.2%
**Awake but motionless (ABM)**	**3.1%**	**1.2%**	**6.8%**	**0.0%**	**20.4%**
Sniffing	1.2%	0.6%	2.5%	0.0%	8.0%
Interacting with object	0.6%	0.0%	1.9%	0.0%	4.9%
Sleeping	0.6%	0.0%	2.8%	0.0%	13.6%
Barking	0.6%	0.0%	4.0%	0.0%	16.7%
Jumping	0.6%	0.0%	2.5%	0.0%	21.0%
Eating	0.6%	0.0%	1.2%	0.0%	3.1%
Grooming	0.6%	0.0%	1.2%	0.0%	4.3%
Interacting with person	0.6%	0.0%	0.6%	0.0%	2.5%
Whining	0.0%	0.0%	0.6%	0.0%	17.9%
Anormal behaviour	0.0%	0.0%	0.9%	0.0%	17.9%
Howl	0.0%	0.0%	0.0%	0.0%	8.6%
Drinking	0.0%	0.0%	0.0%	0.0%	2.5%
Urinating-excreting	0.0%	0.0%	0.6%	0.0%	1.2%

**Table 4 animals-09-00420-t004:** Results of univariate general linear regression models comparing variables from the Kong^TM^ anhedonia test to time spent awake but motionless in the home kennel (logarithmically transformed). N = 43 dogs. Average bout length is mean duration of bouts of interaction with the Kong^TM^. The overall model was significant at the *p* < 0.05 level: F = 3.66, df = 2, *p* = 0.035, R^2^ = 15.8%. See Appendix A for model statistics from all tests. The 95% CI Bounds indicate the upper and lower 2.5% confidence intervals around the beta (B) estimate.

	B	t	*p*	95% CI Bounds
Constant	0.441	4.60	<0.001	0.247 to 0.635
Average bout length	0.001	1.98	0.055	0.000 to 0.002
Weeks in shelter	0.015	2.04	0.048	0.000 to 0.029

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
