# Peer review of "Could Greater Time Spent Displaying Waking Inactivity in the Home Environment Be a Marker for a Depression-Like State in the Domestic Dog?"

_animals, 2019, doi:10.3390/ani9070420_

Round 1

Reviewer 1 Report

This is an interesting and well conducted study that is an important contribution to this field of research. Overall it is also very well written.

Specific Comments

Simple Summary

Line 17: should read ‘One hallmark’ and not ‘hallmarks’.

I believe that the p=0.095 result for the average length of Kong bout use and ABM should not be discussed in as much detail as it is, since it is a non-significant result.

The hypothesis that ABM reflects a depression-like syndrome is not supported only insofar as you assume that the Kong test that you have used will accurately measure depression in dogs. I would not support this, and you do include this limitation in your Discussion.

Introduction

Line 54: ‘However, to date, no empirical evidence has been collected specifically to investigate the possibility that dogs could display depression-like syndromes.’ I would dispute this statement given the early work by Seligman and colleagues looking at learned helplessness in a dog model, which was compared it to depression in humans. One of the interesting parts of this research was that not all dogs displayed learned helplessness under the same treatment conditions.

Line 110: Should read ‘However there are currently a lack of methods…’

Methods

How were the breeds identified-by the owners or by visual inspection? Either way the data may be biased as people are likely to over-estimate the numbers of Staffy cross dogs by appearance alone.

How clear was the difference between the ABM and Lying down head or ears mobile ethogram behaviours? Do you think that if there were only very subtle movements of the head or ears that there might be overlap in these categories? I also have a question of interpretation here- if the dog is both depressed and anxious (which is a potential combination) might you see more of the Lyind down head or ears mobile behaviour as the dog is likely to be more vigilant?

Statistical analysis- I understand that each variable that might affect time spent ABM was analysed independently? What if there were interactions between factors, such as neuter status and age?

Results

In the Methods it is stated that Time spent ABM was not normally distributed, and probably the other behaviours in Table 3 were not either. It would make more sense to provide median rather than mean values.

Discussion

Line 438: I’m not sure that you can summise that the most inactive dogs tended to display an increased interest in the Kong toy- the only parameter that approached significance was the length of bouts, but total time spent interacting with the kong did not. Also you mean the ABM rather than inactive here, as the dogs lying down but moving their head or ears were still inactive, as were the dogs showing sleeping behaviour?

Another interpretation of the results with the dogs is that some animals are active responders and others are passive responders to a stressor. The dogs that you are measuring as ABM might be the passive responders, and more likely to show depressive like symptoms. In the early Seligman studies not all dogs ended up showing learning helplessness, as noted above.

The discussions around the limitations of the study are excellent.

Author Response

This is an interesting and well conducted study that is an important contribution to this field of research. Overall it is also very well written.

Thank you,

Specific Comments

Simple Summary

Line 17: should read ‘One hallmark’ and not ‘hallmarks’.

Oops, sorry. Corrected [line 17].

I believe that the p=0.095 result for the average length of Kong bout use and ABM should not be discussed in as much detail as it is, since it is a non-significant result.

This isn’t relevant any more, please see below for an update on the statistical analysis methods and new result.

The hypothesis that ABM reflects a depression-like syndrome is not supported only insofar as you assume that the Kong test that you have used will accurately measure depression in dogs. I would not support this, and you do include this limitation in your Discussion.

Agreed (as discussed).

Introduction

Line 54: ‘However, to date, no empirical evidence has been collected specifically to investigate the possibility that dogs could display depression-like syndromes.’ I would dispute this statement given the early work by Seligman and colleagues looking at learned helplessness in a dog model, which was compared it to depression in humans. One of the interesting parts of this research was that not all dogs displayed learned helplessness under the same treatment conditions.

Sorry we expressed ourselves poorly before; we have clarified [lines 55-57].

Line 110: Should read ‘However there are currently a lack of methods…’

Oops, thank you. Corrected [line 114].

Methods

How were the breeds identified-by the owners or by visual inspection? Either way the data may be biased as people are likely to over-estimate the numbers of Staffy cross dogs by appearance alone.

Breed information was provided by the shelters and obtained either from pedigrees or by visual inspection. We have added that information and acknowledged the risk for over-estimation of Staffordshire Bull Terrier crosses [lines 209-211]

How clear was the difference between the ABM and Lying down head or ears mobile ethogram behaviours? Do you think that if there were only very subtle movements of the head or ears that there might be overlap in these categories?

We have clarified which behaviour should be coded for in the case of short, subtle head/ear movements happening [Table 2, see ‘Lying down head or ears mobile’ definition]. The good inter-observer agreement [lines 257-258] testify to the difference between ABM and Lying down head or ears mobile being clear to the observers.

I also have a question of interpretation here- if the dog is both depressed and anxious (which is a potential combination) might you see more of the Lyind down head or ears mobile behaviour as the dog is likely to be more vigilant?

We agree that depression and anxiety can be co-morbid in human beings, and that such co-morbidity might be hypothesised to happen in dogs too (a hypothesis that we cannot test in the current study however due to the absence of specific anxiety-related measurements). We also agree that dogs in anxious states are likely to be more vigilant. We are however a little confused (sorry if we missed the point here) by the suggestion that, in case of co-morbid anxiety, we might see more ‘Lying down head or ears mobile’. This is because we would expect vigilant behaviour in dogs to more typically be associated with increased activity (e.g. increased locomotion and observation near the end of the kennel, barking etc.), and as such we are not sure we would predict to observe more lying down head or ears mobile, which would better fit into a classification of ‘resting’ rather than ‘vigilant’. Whilst clear definitions of ‘vigilance’ are often lacking in the literature, this study by Estellés & Mills (https://veterinaryrecord.bmj.com/content/159/5/143) found vigilance to be associated with barking, jumping, and restlessness.

Statistical analysis- I understand that each variable that might affect time spent ABM was analysed independently? What if there were interactions between factors, such as neuter status and age?

Thank you for pointing this out. It was quite uncanny timing actually as an hour before your review was sent through to us I was reading a paper from 1996 highlighting the pitfalls of the usual methods for screening independent variables (‘Inappropriate Use of Bivariable Analysis to Screen Risk Factors for Use in Multivariable Analysis’) and it got me thinking about this very point. In light of this paper, and your comment, we have now re-analysed the association between the Kong variables and logABM including each fixed factor that could potentially be a confounder for the Kong variables (previously I had mistakenly only been considering confounders for ABM). We tested for correlations (using Kendall’s tau-b in case of ties) between each of these variables (sex, neuter status, weeks in shelter, age, weight and breed group) to decide which may need to be included together. Significant correlations ranging from -0.42** to 0.57*** were found between each of the three variables weight, breed group and sex, so these variables were included together in all models (which dropped the sample size for these models to 38 as we did not have weight for all dogs), however they had no impact on any models. There were no other associations between any other factors. This process and the results [process: lines 355-360; results: lines 443-452 and Table 4] are now fully described and two new supplementary tables have been added (Table S3 & S4). Only one significant multi-variable model (p=0.035) was formed from this process, which combined the factors ‘weeks in shelter’ and ‘average bout length’, effectively controlling for a confounding effect of  time in the shelter and improving the effect and significance of ‘average bout length’, which previously when included alone achieved a model with a significance of p=0.095. Essentially, this result shows a stratifying effect of weeks in shelter, with ‘average bout length’ becoming associated with ABM only when dogs have been in the shelter for longer. We have altered the Discussion [lines 521-523] and Results accordingly.

Results

In the Methods it is stated that Time spent ABM was not normally distributed, and probably the other behaviours in Table 3 were not either. It would make more sense to provide median rather than mean values.

Thank you for pointing this out. We have replaced means and SD by median and 1st and 3rd quartiles wherever relevant in the text, both for time spent ABM and Kong (not normally distributed either) values [main manuscript: lines 373-379, lines 418-423, Table 3; supplementary material: lines 88-92, 102-103, Table S1]

Discussion

Line 438: I’m not sure that you can summise that the most inactive dogs tended to display an increased interest in the Kong toy- the only parameter that approached significance was the length of bouts, but total time spent interacting with the kong did not. Edited [lines 542-544]

Also you mean the ABM rather than inactive here, as the dogs lying down but moving their head or ears were still inactive, as were the dogs showing sleeping behaviour? Yes sorry, we meant ABM indeed, now corrected [line 533]. We have spotted a similar confusing wording somewhere else in the discussion, also corrected [line 543-543].

Another interpretation of the results with the dogs is that some animals are active responders and others are passive responders to a stressor. The dogs that you are measuring as ABM might be the passive responders, and more likely to show depressive like symptoms. In the early Seligman studies not all dogs ended up showing learning helplessness, as noted above.

We have edited the discussion accordingly [lines 599-503]

The discussions around the limitations of the study are excellent.

Thank you, and thank you for your comments and for taking the time to review our manuscript.

Reviewer 2 Report

I really enjoyed reading this manuscript, I thought it was well-written and thought out. I fully appreciate including the pilot study information as it helped immensely in my understanding of the thought process of the authors. Certainly, I think the topic of interest is interesting and the results bring something relevant to the table. However, I have a few concerns that the authors may or may not be able to address.

Line 169 - It is stated that the dogs were walked "at least twice a day or 10-20 minutes" but at some shelters was this significantly different? I don't need to note that differences in exercise can affect many of the variables investigated in this study, thus showing a minimum and not a range or average +/- is not acceptable here.

2.1.2 I am interested in why the number of dogs in this qualitative behavior analyses is so low compared to t he rest of the study - for example, only 3 strays dogs were included, this is a small number and at least the reasoning should be addressed, if they cannot include more dogs.

216 (and others) I am concerned about the instantaneous sampling used here, though including the supplementary material was very helpful here. How can the authors be sure that the 30-second interval sampling is already inclusive enough to include as the "baseline" to compare to 30s, 1, 1.5 and 2 minute sampling? From here, I am not sure what is lost going from continuous to 30 second sampling, and it does not look like the authors looked into that. This would be important, especially because they are looking at state behaviors. I fully appreciate trying to reduce time spent coding, and I am willing to be convinced, but I am not quite there. Regardless, I would put more direct wording into the manuscript that this sampling was validated (somewhat) with the pilot study.

Line 242 - It is stated that agreement was checked and rechecked during the pilot study, however I find no mention of it in the pilot study/supplementary material. Please add how agreement was addressed statistically and at what points.

Line 245 - Loose or lose?

Line 260-261 - "it has been shown that dogs treat filled Kong's as feeding devised as opposed to toys" - dogs liked? preferred?

Line 272 - Can authors address how kongs were kept filled, effectively, the same, and not just by weight. For example, I assume that the dogs' "kibble" holds a relatively lower value to them than the "KONG Stuff'n paste"

Line 288 - asses or assess?

Line 369 - I don't appreciate the statement "was approaching significance at p<0.05" as it appears the authors are referring to the result of p=0.095 found in table 4 (and it is not mentioned nor found in the text, unless readers then look to the table) - I would leave it up to the editor to decide if this is appropriate. 

Line 385 - affected?

I would like to note that I especially enjoyed the discussion, I felt it was appropriate for the results found and honest where the authors felt that the study could either be improved upon or expanded. I have no comments, it is well-written and appropriately encompassing.

Author Response

I really enjoyed reading this manuscript, I thought it was well-written and thought out. I fully appreciate including the pilot study information as it helped immensely in my understanding of the thought process of the authors. Certainly, I think the topic of interest is interesting and the results bring something relevant to the table. However, I have a few concerns that the authors may or may not be able to address.

Thank you, and we hope the revised version of the manuscript addresses your concerns.

Line 169 - It is stated that the dogs were walked "at least twice a day or 10-20 minutes" but at some shelters was this significantly different? I don't need to note that differences in exercise can affect many of the variables investigated in this study, thus showing a minimum and not a range or average +/- is not acceptable here.

Sorry we expressed ourselves poorly before; walking routine was similar across shelters, we have clarified [lines 174-175].

2.1.2 I am interested in why the number of dogs in this qualitative behavior analyses is so low compared to the rest of the study - for example, only 3 strays dogs were included, this is a small number and at least the reasoning should be addressed, if they cannot include more dogs.

We are afraid the reasons here are purely practical: we only had data from the 14 pilot study dogs at the time we could organise the QBA study, while once the full study sample data were collected, we had neither time nor money left to organise another QBA assessment. We have clarified in the manuscript [lines 193-195].  

216 (and others) I am concerned about the instantaneous sampling used here, though including the supplementary material was very helpful here. How can the authors be sure that the 30-second interval sampling is already inclusive enough to include as the "baseline" to compare to 30s, 1, 1.5 and 2 minute sampling? From here, I am not sure what is lost going from continuous to 30 second sampling, and it does not look like the authors looked into that. This would be important, especially because they are looking at state behaviors. I fully appreciate trying to reduce time spent coding, and I am willing to be convinced, but I am not quite there. Regardless, I would put more direct wording into the manuscript that this sampling was validated (somewhat) with the pilot study.

We appreciate this comment and agree that further description should be included the main text, now added [lines 263-275]. We also agree that comparing our tested sampling period against continuous sampling (instead of 30s as ‘baseline’) would have been even better. Unfortunately recording continuously did take the intern longer than scanning at 30s and simply was not possible given our limited resources. We would however like to emphasize that because several behaviours (including the ABM behaviour related to hypothesis under test) were state behaviours, 5 seconds were watched continuously either side of each scan point (in other words, while scanning at 30s interval, 25% of each minute, or 1h out of the 4 hours, were also recorded continuously). This should give a good representation of time budget, even if we acknowledge that it does not exclude the possibility that short periods of ABM were missed. We would also like to emphasize that even though we had to compromise (this potential reduction in estimate accuracy was an unfortunate but necessary trade-off against limited resources), we feel like we have already gone beyond the norm about justifying our sampling period selection.

Line 242 - It is stated that agreement was checked and rechecked during the pilot study, however I find no mention of it in the pilot study/supplementary material. Please add how agreement was addressed statistically and at what points.

Good point, thank you for spotting that – clarified [lines 257-258]

Line 245 - Loose or lose?

Oops, lose sorry. Corrected [line 260]

Line 260-261 - "it has been shown that dogs treat filled Kong's as feeding devised as opposed to toys" - dogs liked? preferred?

Apologies, we expressed ourselves poorly before; we just meant that dogs habituated to Kong’s display behaviours showing that they expect food from it, not just playing with the KongTM as with other rubber toys; clarified [lines 300-302]

Line 272 - Can authors address how kongs were kept filled, effectively, the same, and not just by weight. For example, I assume that the dogs' "kibble" holds a relatively lower value to them than the "KONG Stuff'n paste"

Good point, thank you for spotting that – clarified [lines 285-287]

Line 288 - asses or assess?

Oops, assess, thank you. Corrected [line 328]

Line 369 - I don't appreciate the statement "was approaching significance at p<0.05" as it appears the authors are referring to the result of p=0.095 found in table 4 (and it is not mentioned nor found in the text, unless readers then look to the table) - I would leave it up to the editor to decide if this is appropriate. 

This is no longer relevant [reviewer 1 advised on conducting new statistical analyses, see: lines 355-360 for description of the process and lines 443-451 and Table 4, Table S3 and Table S4 for updated results].

Line 385 - affected?

Oops, yes affected sorry. Corrected [line 467]

I would like to note that I especially enjoyed the discussion, I felt it was appropriate for the results found and honest where the authors felt that the study could either be improved upon or expanded. I have no comments, it is well-written and appropriately encompassing.

Thank you, and thank you for your comments and for taking the time to review our manuscript.

Round 2

Reviewer 2 Report

All of my comments were addressed adequately - I would consider this acceptable for publication in present form.